# The proton channel OTOP1 is a sensor for the taste of ammonium chloride

Ziyu Liang [1,2,5], Courtney E. Wilson[3,5], Bochuan Teng[1,2,4], Sue C. Kinnamon[3] & Emily R. Liman [1] ✉

Ammonium ($NH_4^+$), a breakdown product of amino acids that can be toxic at high levels, is detected by taste systems of organisms ranging from *C. elegans* to humans and has been used for decades in vertebrate taste research. Here we report that OTOP1, a proton-selective ion channel expressed in sour (Type III) taste receptor cells (TRCs), functions as sensor for ammonium chloride ($NH_4Cl$). Extracellular $NH_4Cl$ evoked large dose-dependent inward currents in HEK-293 cells expressing murine OTOP1 (mOTOP1), human OTOP1 and other species variants of OTOP1, that correlated with its ability to alkalinize the cell cytosol. Mutation of a conserved intracellular arginine residue (R292) in the mOTOP1 tm 6-tm 7 linker specifically decreased responses to $NH_4Cl$ relative to acid stimuli. Taste responses to $NH_4Cl$ measured from isolated Type III TRCs, or gustatory nerves were strongly attenuated or eliminated in an *Otop1*$^{-/-}$ mouse strain. Behavioral aversion of mice to $NH_4Cl$, reduced in *Skn-1a*$^{-/-}$ mice lacking Type II TRCs, was entirely abolished in a double knockout with *Otop1*. These data together reveal an unexpected role for the proton channel OTOP1 in mediating a major component of the taste of $NH_4Cl$ and a previously undescribed channel activation mechanism.

Ammonium ($NH_4^+$) and its gas, ammonia ($NH_3$), breakdown products of amino acids, are generally noxious to humans and other animals. High tissue concentrations of ammonium/ammonia found in conditions such as hyperammonemia can be life-threatening[1,2]. Thus, it is not surprising that animals from the nematode *C. elegans* to fruit flies and humans have evolved multiple mechanisms to detect ammonium/ammonia[3,4]. Ammonium has a unique and strong taste, described as a combination of bitter, salty, and a little sour[5]. And although generally aversive, humans can learn to enjoy the taste of $NH_4Cl$ which is added to a confection that is popular in Scandinavian countries called salty licorice. Because it is such a strong gustatory stimulus, $NH_4Cl$ has been used for decades as a control stimulus and to normalize data when performing recordings from gustatory nerves that innervate the tongue[6–8]. Yet the mechanism by which ammonium activates taste cells is poorly understood.

Taste buds are composed of 50–100 taste receptor cells (TRCs), which are specialized epithelial-derived cells that communicate to gustatory nerve fibers with cell bodies in the geniculate and petrosal ganglia[9,10]. The Type II TRCs each detect bitter, sweet, or umami, and signal through the ion channel TRPM5[11–13], while Type III TRCs sense acid stimuli (sour taste) through the proton channel OTOP1[14–17]. Yet a third cell type mediates the attractive taste of NaCl using an ENaC channel[18,19]. Early experiments showed that single gustatory nerve fibers sensitive to HCl also responded to $NH_4Cl$[20,21], which was the first evidence that the Type III TRCs might participate in the detection of $NH_4Cl$.

Ammonium in solution is in equilibrium with ammonia, which acts like a Brønsted-Lowry base, according to the reaction: $H_3O^+ + NH_3 <-> NH_4^+ + H_2O$. The $NH_3$ gas crosses the cell membranes, and once within the cytosol regenerates $NH_4^+$, thereby

[1]Section of Neurobiology, Department of Biological Sciences, University of Southern California, Los Angeles, CA 90089, USA. [2]Program in Neuroscience, University of Southern California, Los Angeles, CA 90089, USA. [3]Department of Otolaryngology, University of Colorado Medical School, 12700 E 19(th) Avenue, MS 8606, Aurora, CO 80045, USA. [4]Present address: Division of Biology and Biological Engineering, California Institute of Technology, Pasadena, CA, USA. [5]These authors contributed equally: Ziyu Liang, Courtney E. Wilson. ✉e-mail: liman@usc.edu

reducing the concentration of intracellular H[+] and alkalinizing the cell cytosol[22,23]. The effect of NH₄Cl to alkalinize the cell cytosol is nearly ubiquitous and has been used over the decades to study pH regulation within diverse cell types[24,25]. The known effect of NH₄Cl to raise intracellular pH, thereby generating a driving force for proton entry, suggests that the sensor for NH₄Cl could be a proton channel.

Here we set out to test the contribution of Type III TRCs and OTOP1 to sensory responses of vertebrates to NH₄Cl. We show that OTOP1 is activated by NH₄Cl in heterologous cells and is required for the sensory responses in isolated murine Type III TRCs. We also find that OTOP1 is required for the gustatory nerve response to NH₄Cl and that in a mouse missing Type II TRCs, behavioral responses to NH₄Cl are entirely dependent on a functional OTOP1 channel. Finally, we find that activation of NH₄Cl is conserved across species variants of OTOP1 and involves regulation of the channel by a titratable residue on the intracellular surface of the protein.

## Results

### Type III (sour) taste receptor cells mediate a component of the gustatory response to NH₄Cl

We first set out to assess the contribution of Type III acid-sensing taste receptor cells (sour cells) to the taste of NH₄Cl by recording responses in vivo from the chorda tympani (CT) nerve which innervates taste buds at the front of the tongue. As first documented decades ago, 100 mM NH₄Cl, at near neutral pH, applied to the anterior tongue evoked a large response from the chorda tympani nerve of wild-type mice (Fig. 1a). We next tested if the response to NH₄Cl was maintained in an *Skn-1a* (also known as *Pou2f3*) knockout mouse strain that retains Type III TRCs but lacks taste receptor cells that mediate bitter, sweet, umami, and amiloride-sensitive sodium taste[26,27]. Chorda tympani nerve responses to NH₄Cl in the *Skn-1a⁻/⁻* mice were robust and not significantly different from responses observed in wild-type mice (Fig. 1a, b). Responses to the sweet compound AceK were strongly attenuated in *Skn-1a⁻/⁻* compared with wild-type mice, as previously demonstrated (Fig. 1b). The *Skn-1a⁻/⁻* mice retained sensitivity to citric

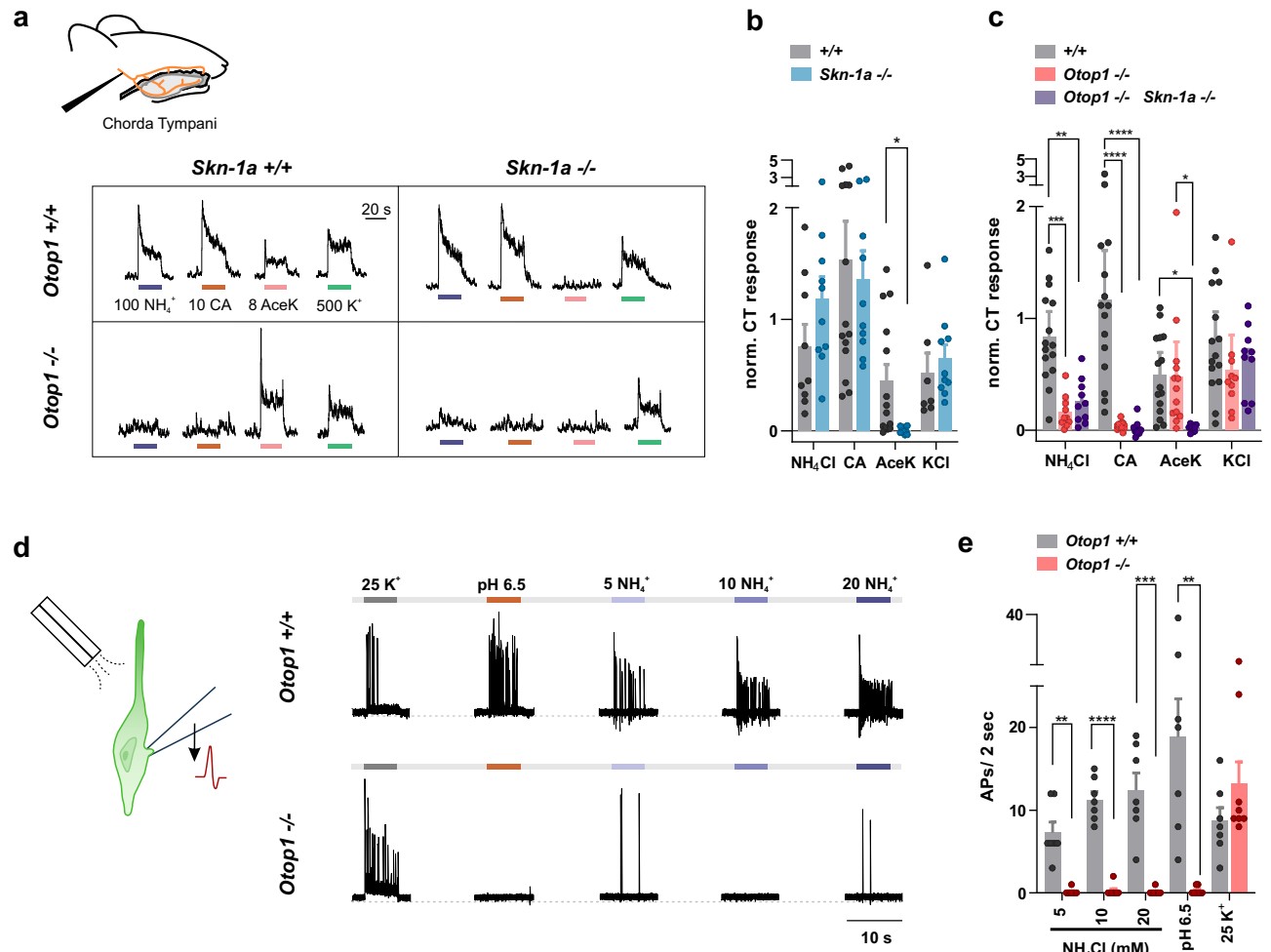

**Fig. 1 | Type III TRCs respond to ammonium chloride in an OTOP1-dependent manner. a** Representative traces of chorda tympani (CT) responses from wild-type (*Otop1⁺/⁺ Skn-1a⁺/⁺*), *Otop1⁻/⁻*, *Skn-1a⁻/⁻*, and *Otop1⁻/⁻ Skn-1a⁻/⁻* double knockout mice to application to the anterior tongue of 100 mM NH₄Cl, 10 mM citric acid (CA), 8 mM AceK and 500 mM KCl. **b, c** Average data (mean ± SEM) and scatterplot of normalized CT response from mouse strains as indicated in response to taste stimuli from experiments as in **a**. Statistics significance was determined by two-way ANOVA with Bonferonni correction for multiple comparisons. *$p < 0.05$, **$p < 0.01$, ***$p < 0.001$. ****$p < 0.001$. **b** $n = 12$ wild-type mice, $n = 10$ *Skn-1a⁻/⁻* mice. **c** $n = 15$ wild-

type mice, $n = 13$ *Otop1⁻/⁻* mice, $n = 10$ double knockout mice. **d** Action potentials were measured in cell-attached recordings from isolated YFP⁺ Type III TRCs from wild-type and *Otop1⁻/⁻* mice in response to extracellular stimuli as indicated (concentrations in mM). **e** Number of action potentials measured in the first 2 s in response to various stimuli from experiments as in **d**. Data from individual cells and the mean ± SEM are shown. Significance comparisons of wild-type and *Otop1⁻/⁻* TRCs were determined by two-tailed Student's *t* test. (all comparisons between the *Otop1⁻/⁻* and wild-type were significant at $p < 0.0001$ except high K⁺ for which $p = 0.174101$). $n = 7$ cells for wild-type, and $n = 8$ cells for *Otop1⁻/⁻*.

acid and to a high concentration of KCl, previously shown to activate both Type II and Type III TRCs[28,29] (Fig. 1a, b). These data demonstrate that the remaining taste receptor cells in the *Skn-1a*[−/−] mice, the Type III TRCs, are sufficient to mediate gustatory nerve responses to NH$_4$Cl.

To directly test if Type III TRCs can be activated by NH$_4$Cl, we used cell-attached patch-clamp recording to measure action potentials in acutely isolated taste receptor cells. Type III TRCs were identified based on the expression of YFP from the *Pkd2l1* promoter, a specific marker for these cells[30–33]. Indeed, NH$_4$Cl elicited robust trains of action potentials starting at a concentration as low as 5 mM and the firing rate increased in a dose-dependent manner up to the highest concentration of NH$_4$Cl tested, 20 mM (Fig. 1d, e). An acid stimulus (pH 6.5) and 25 mM KCl, both of which elicited strong activation, served as positive controls.

Together these data support the conclusion that Type III TRCs mediate a component of the gustatory response to NH$_4$Cl.

## OTOP1 is essential for the sensory response of Type III TRCs to NH$_4$Cl

Type III TRCs express the *Otop1* gene that encodes a proton-selective ion channel (OTOP1)[14]. NH$_4$Cl applied at neutral pH is expected to increase intracellular pH, creating a driving force for proton entry (see below). We, therefore, considered that as a proton channel, OTOP1 was a good candidate to function as a receptor for NH$_4$Cl. We tested whether gustatory nerve responses and responses of isolated TRCs were mediated by OTOP1 by repeating the experiments described above using a mouse strain carrying an inactivating mutation in the *Otop1* gene (*Otop1*[−/−])[16]. Type III TRCs from *Otop1*[−/−] mice were insensitive and did not generate action potentials in response to NH$_4$Cl over a concentration range that elicited robust responses in wild-type cells (Fig. 1d, e). Responses to high potassium were retained, confirming that the cells were healthy and electrically excitable. Similarly, we found that responses of the chorda tympani nerve to NH$_4$Cl depended on a functional *Otop1* gene. Chorda tympani responses to NH$_4$Cl were strongly attenuated in the *Otop1*[−/−] mouse strain, which retained sensitivity to the sweet stimulus AceK, used as a control (Fig. 1a, c). A double knockout of *Otop1* and *Skn-1a* showed a similarly attenuated response to NH$_4$Cl as the *Otop1*[−/−]-strain (Fig. 1a, c); these mice also lost sensitivity to AceK, but retained sensitivity to high potassium, which was used as a control for the health of the nerve (Fig. 1a, c). The *Otop1*[−/−] mice in either a wild-type or *Skn-1a*[−/−] background also showed a severely attenuated response to citric acid, as expected.

Zn$^{2+}$ has been shown to inhibit OTOP1 currents in a dose- and pH-dependent manner[14,16,33,34]. If OTOP1 functions as the receptor, we expected that extracellular Zn$^{2+}$ would inhibit NH$_4$Cl-evoked taste responses measured both from the CT nerve in intact animals and from isolated Type III TRCs. Indeed, concentrations of Zn$^{2+}$ from 1–10 mM inhibited the CT nerve response to 100 mM NH$_4$Cl in a dose-dependent manner, with ~70% inhibition observed at a concentration of 10 mM (Fig. 2a, b). Similarly, action potentials measured from isolated Type III TRCs in response to 10 mM NH$_4$Cl were inhibited by 1 mM Zn$^{2+}$ (Fig. 2c, d). Inhibition of NH$_4$Cl evoked responses by Zn$^{2+}$ in isolated cells was dose-dependent starting at a concentration as low as 1 μM and saturating at 10 μM (Fig. 2e, f). The difference in sensitivity of the response from isolated cells and the chorda tympani nerve suggests a dilution of the stimulus as it reaches the surface of the taste cells in the intact tissue by a factor of ~1000.

Together these data show that OTOP1 channels in Type III TRCs mediate the excitatory effects of NH$_4$Cl measured both in isolated cells and in recordings from the chorda tympani nerves of intact animals.

## OTOP1 carries large inward currents in response to NH$_4$Cl

One expectation, if the ion channel OTOP1 functions as a sensor for NH$_4$Cl, is that it should transduce stimulation with NH$_4$Cl into an electrical response. Specifically, we hypothesized that if NH$_4$Cl acts

directly on OTOP1, we should be able to observe its activating effects on OTOP1 expressed in a heterologous cell type. Indeed, in HEK-293 cells transfected with murine *Otop1* (*mOtop1*), NH$_4$Cl evoked large inward currents (Fig. 3a, b). Inward currents increased in magnitude in a dose-dependent manner over a range of 1–160 mM NH$_4$Cl with saturation at ~80 mM NH$_4$Cl. Under the same conditions, no currents were observed in un-transfected cells. Somewhat surprisingly, the currents elicited in response to the higher concentrations of NH$_4$Cl were comparable in magnitude to those elicited by an acid stimulus at pH 5.5 (Fig. 3c). This finding suggests that NH$_4$Cl may be an important and physiologically relevant activator of the OTOP1 channel.

We initially hypothesized that the response of OTOP1 to NH$_4$Cl was a consequence of its ability to produce intracellular alkalization, creating a driving force for proton entry through the channels. To directly test this hypothesis, we co-transfected *mOtop1* with cDNA encoding pHlourin, a pH-sensitive variant of GFP[35]. Imaging of the fluorescent emission of pHlourin while performing patch-clamp recording showed that NH$_4$Cl induced an increase in intracellular pH that paralleled its ability to evoke OTOP1 currents (Fig. 3d, e). At low concentrations of NH$_4$Cl (5 mM), the pH and current responses showed similar kinetics while at higher concentrations, the currents continued to increase modestly even when the pH signal had plateaued (Fig. 3e).

To estimate the change in intracellular pH that was produced as a response to the application of NH$_4$Cl, we measured the reversal potential of the evoked currents during a ramp depolarization and from this calculated pH$_i$ (Fig. 3f and methods). This showed that in response to 160 mM NH$_4$Cl, the intracellular pH reached a value of ~pH 8.2, which is consistent with other reports on the effects of NH$_4$Cl[23].

We also confirmed that mOTOP1 currents elicited in response to NH$_4$Cl could be inhibited by Zn$^{2+}$. Application of extracellular Zn$^{2+}$ inhibited NH$_4$Cl evoked currents in a dose-dependent manner with an IC$_{50}$ of 1.25 μM (Supplementary Fig. 1). This is similar to the inhibition of NH$_4$Cl action potentials in isolated taste cells (see above). The higher affinity of the channel for Zn$^{2+}$ when activated at neutral versus acidic pH (IC$_{50}$ = 0.20 mM, pH 5.5) is expected from its known pH dependence[34].

Together these data show that NH$_4$Cl elicits large mOTOP1-dependent inward currents, sensitive to inhibition by Zn$^{2+}$, consistent with a role for mOTOP1 as the sensor for NH$_4$Cl in Type III TRCs.

## Sensitivity to NH$_4$Cl is conserved across species variants of OTOP1

We next asked if sensitivity to NH$_4$Cl is conserved among different species variants of OTOP channels, as might be expected given the broad range of organisms that find the taste of NH$_4$Cl aversive. When expressed in HEK-293 cells, human, chicken, and zebrafish OTOP1 channels all responded to NH$_4$Cl with large inward currents (Fig. 4a); in each case, the ammonium-induced currents were inhibited by 1 mM Zn$^{2+}$. For mouse OTOP1, we observed potentiation of the response to NH$_4$Cl following the removal of Zn$^{2+}$, as previously reported for mild acid-evoked responses[36]. Interestingly, we observed variation in the magnitudes of the responses to NH$_4$Cl among the different species variants of OTOP1 (Fig. 4b). For example, relative to its response to the acidic stimulus, chicken OTOP1 carried much larger inward currents in response to NH$_4$Cl. Meanwhile, zebrafish OTOP1 was less sensitive to NH$_4$Cl but responded well to the acid stimulus. Human OTOP1 was intermediate, responding well to both the acid stimulus and NH$_4$Cl but with somewhat different kinetics than mouse OTOP1. Thus, in all species tested, and likely all vertebrate species, OTOP1 could serve as a sensor for NH$_4$Cl. Differences in the sensitivity of the species variants of OTOP1 to NH$_4$Cl and acids may reflect a diversity in behavioral ecology.

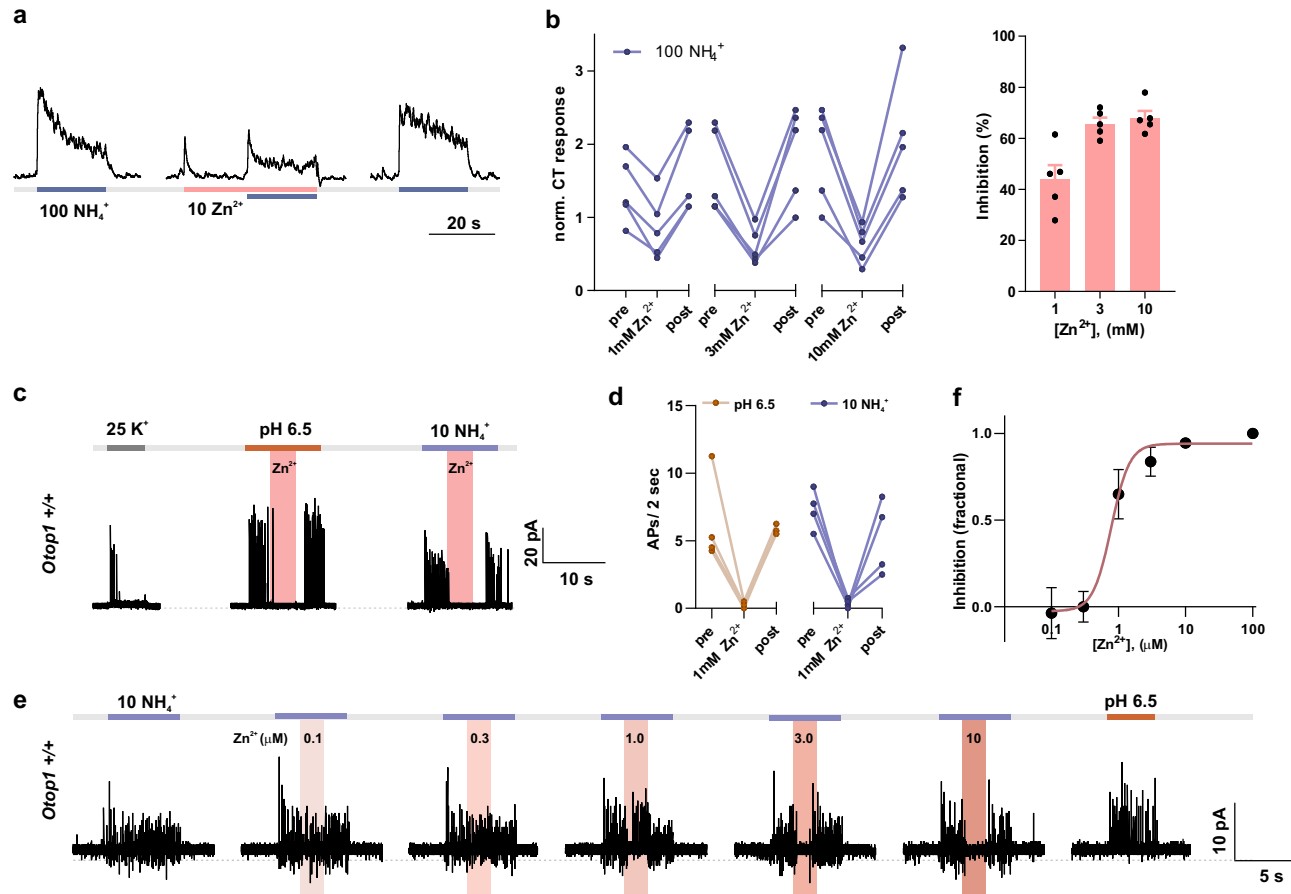

**Fig. 2 | Zn²⁺ inhibits the NH₄Cl-evoked responses in chorda tympani nerve recording and isolated TRCs. a** Chorda tympani response measured from wild-type mice in response to extracellular 100 mM NH₄Cl with and without 10 mM Zn²⁺. **b** Left panel, normalized CT response measured from the experiment as in **a**. Right panel, average data (mean ± SEM, $n = 5$ mice), and scatterplot of the fractional inhibition of the CT response as a function of extracellular Zn²⁺ concentration. **c** Action potentials were measured from isolated TRCs of wild-type mice in response to extracellular 25 mM K⁺, pH 6.5, and 10 mM NH₄⁺, together with 1 mM

Zn²⁺ applied at the times indicated. **d** Number of action potentials measured in the first 2 s after the stimulus change from experiments as in **c**. Data from individual cells are shown with connected lines ($n = 4$). **e** Dose-dependent inhibition of the NH₄⁺ induced action potentials by Zn²⁺ (pink bar, concentration indicated in μM) measured from isolated TRCs of wild-type mice. **f** Fractional inhibition measured from data (mean ± SEM) as in **e** fit with the Hill equation with an IC₅₀ of 0.78 μM and Hill coefficient of 3.2 ($n = 4$ cells).

## Role of a conserved intracellular arginine residue in activation of mOTOP1 by NH₄Cl

While we initially hypothesized that intracellular alkalization and the change in driving force on the proton could be sufficient to promote proton influx through OTOP1 channels, several pieces of evidence suggested that the response of OTOP1 to NH₄Cl might not be entirely passive. Most striking was the similar magnitude of the responses of mOTOP1 to NH₄Cl and the acid stimulus, despite the ten-fold larger H⁺ gradient for the acid stimulus (pH₀ 5.5/pHᵢ 7.4) than for NH₄Cl (pH₀ 7.4/pHᵢ - 8.2). This prompted us to consider whether the OTOP1 channels, which are mostly nonconductive at neutral pH₀[37], might be made more conductive in response to intracellular alkalization secondary to NH₃ diffusion across the cell membrane.

Intracellular alkalization could in principle be sensed by basic residues on the intracellular surface of the channels. We identified four such residues, conserved across a large array of vertebrate OTOP1 channels: K187 (tm 4-5), R292 (tm 6-7), K527, and R528 (tm 10-11) (Fig. 5a). Each residue was mutated to alanine and the mutant channels were tested for sensitivity to NH₄Cl and to an acid stimulus (pH 5.5). One mutation, from arginine to alanine at residue R292 (R292A) greatly reduced the magnitude of the mOTOP1 currents in response to NH₄Cl, while the response to the acid stimulus (pH 5.5) remained almost unchanged (Fig. 5b–d). The reduction of the NH₄Cl-evoked response

was observed over the entire concentration range of 10–160 mM (Fig. 5c, d). We also noted that the kinetics of the residual currents elicited in response to NH₄Cl no longer showed a slow-rising phase, which might reflect a change in the gating of the channel. An arginine at a position equivalent to 292 is present in most vertebrate OTOP channels and some invertebrate OTOP channels (Fig. 5e). Thus, ammonium sensing by OTOP channels may be conserved through evolution.

## Otop1 is required for behavioral responses to NH₄Cl

These data show that, based on responses of heterologously expressed channels, taste receptor cells, and gustatory neurons, OTOP1 functions as a sensor within Type III TRCs for NH₄Cl. If this is indeed true, then the behavioral aversion of mice to the taste of NH₄Cl would be expected to be altered by the targeted deletion of *Otop1*. We tested *Otop1⁻/⁻* mice both in a wild-type background and in a *Skn-1a⁻/⁻* background using a brief access taste test that avoids the confounds of post-ingestive effects (Fig. 6a, b). All taste stimuli were dissolved in artificial saliva, to provide a minimal amount of pH buffering. Wild-type mice find the taste of NH₄Cl strongly aversive, and we observed a sharp decline in licking as concentrations of NH₄Cl were raised above 100 mM. Similarly, *Skn-1a⁻/⁻* mice showed robust aversion to 300 and 500 mM NH₄Cl, although the degree of aversion was reduced

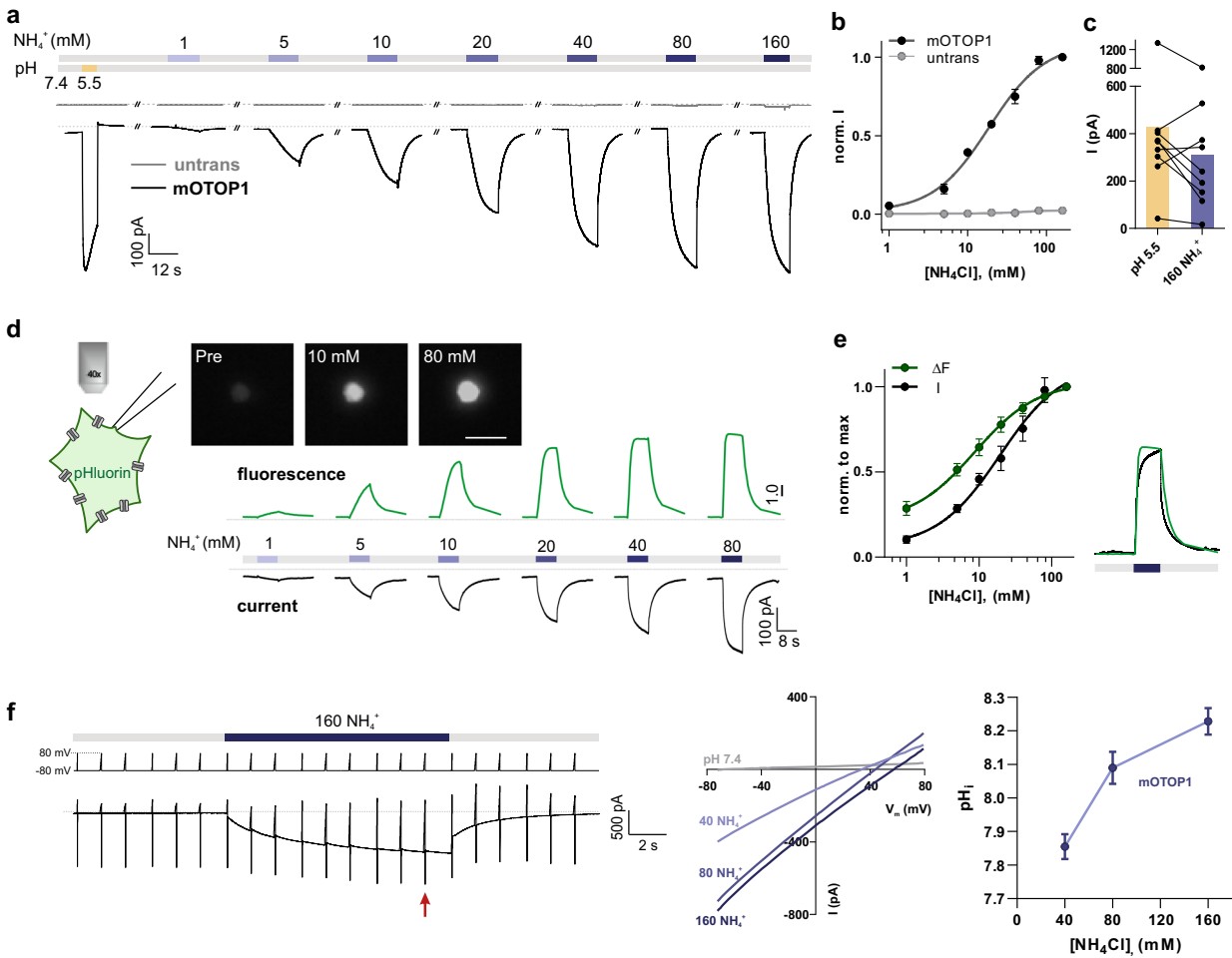

**Fig. 3 | Inward currents and intracellular alkalization in mOTOP1-expressing HEK-293 cells in response to extracellular NH₄Cl.** **a** NH₄Cl induced currents measured with whole-cell patch-clamp recording from HEK-293 cells expressing mOTOP1 or un-transfected cells as labeled ($V_m = -80$ mV). **b** mOTOP1-expressing cells respond to extracellular NH₄Cl in a dose-dependent manner while no current is induced in un-transfected cells. The normalized current magnitude (mean ± SEM) was fit with a Hill slope = 1.23 and $EC_{50} = 18.66$ mM. $n = 7$ mOTOP1-expressing cells, and $n = 8$ un-transfected cells. **c** Current magnitude in response to pH 5.5 and 160 mM NH₄Cl for HEK-293 cells expressing mOTOP1. $n = 9$ (**d**) upper: representative fluorescent images of a cell expressing pHluorin and mOTOP1 captured before and during NH₄⁺ application. Scale bar = 40 μm. lower: representative trace of the current magnitude recorded by whole-cell patch (black) and changes in fluorescent emission (green) upon exposure to varying concentrations of NH₄⁺ from an OTOP1-expressing cell. Data is representative of 6 independent cells and

measurement. **e** Left panel: average data (mean ± SEM, $n = 6$ cells) of responses to varying concentrations of NH₄⁺ (black, current magnitude; green, change in fluorescent) from experiments as in **d**. Results of individual cells were normalized to the responses to 160 mM NH₄⁺. Right panel shows the overlay of the fluorescence with the current from the experiment shown in **d**. **f** Left panel shows the voltage and solution exchange protocol used to measure the reversal potential in response to extracellular 160 mM NH₄⁺. $V_m$ was held at −80 mV and ramped to +80 mV (1 V/s at 1 Hz). The last ramp during the currents peaked was used for later measurements (red arrowhead). Middle panel shows the representative I–V relationship from HEK-293 cells expressing mOTOP1 in response to 40, 80, or 160 mM NH₄⁺ from experiments described in **f** Right panel shows the average estimation (mean ± SEM, $n = 7$ cells) of intracellular pH as a function of extracellular NH₄Cl concentration. The estimated $pH_i$ was calculated from $E_{rev}$ as measured in the middle panel, using the Nernst Equation (see methods).

compared with wild-type mice, indicating that the Type III TRCs are sufficient to mediate a component of the behavioral aversion to NH₄Cl. As controls, we tested the response of *Skn-1a⁻/⁻* mice to a bitter compound (quinine) and a sour stimulus (citric acid) and found, as expected, that the mice were indifferent to even the highest concentration of quinine, and retained sensitivity to acids (Fig. 6c).

We next tested whether OTOP1 contributes to behavioral aversion to NH₄Cl. *Otop1⁻/⁻* mice showed a significantly diminished aversion to NH₄Cl at intermediate concentrations of NH₄Cl (300 mM) (Fig. 6c). Even more remarkably, the double knockout mice of *Skn-1a* and *Otop1* were essentially indifferent to all concentrations of NH₄Cl up to 500 mM NH₄Cl (Fig. 6c). Importantly, the double knockout mice were significantly less sensitive to 500 mM NH₄Cl than *Skn-1a⁻/⁻* mice (Fig. 6c; $p < 0.001$), confirming an essential role for OTOP1 in

mediating a component of the behavioral response to NH₄Cl. All mice retained aversion to citric acid, consistent with earlier reports that OTOP1 and Type III TRCs are dispensable for behavioral aversion to sour tastes[15,16,38].

Thus, we conclude that OTOP1 functions as the NH₄Cl sensor within Type III TRCs and is essential for a component of the behavioral aversion to NH₄Cl mediated by these cells.

## Discussion

The mechanisms by which ammonium/ammonia act on sensory receptors has received considerable attention, both in vertebrate and invertebrates, but as yet, no receptor or signaling mechanism for the taste of NH₄Cl in vertebrates has been identified. Here we show that a major component of the chorda tympani nerve response to NH₄Cl can

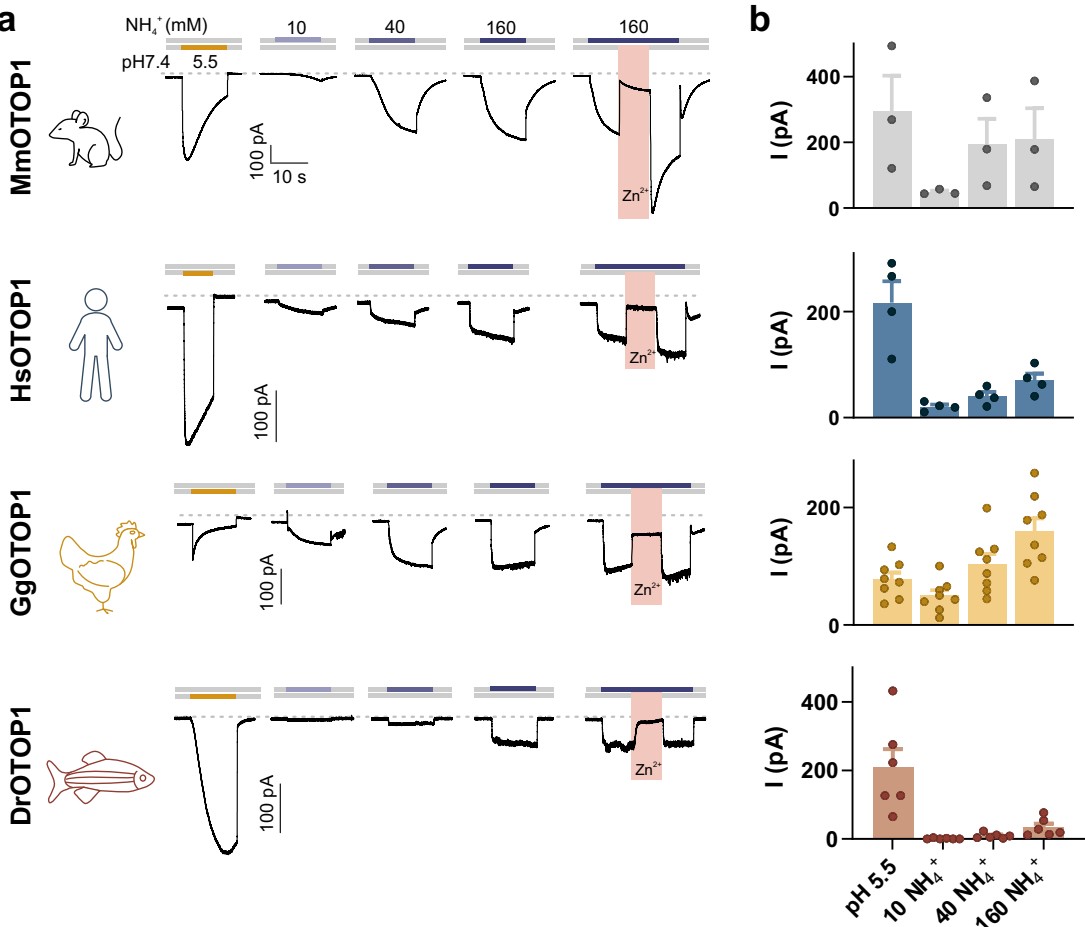

**Fig. 4 | Response to ammonium is evolutionarily conserved among OTOP1 family members from diverse species. a** Representative traces ($V_m = -80$ mV) showing currents evoked in HEK-293 cells expressing mOTOP1 from multiple species including mouse (*Mus musculus*, green), human (*Homo sapiens*, blue), chicken (*Gallus gallus*, yellow) and zebrafish (*Danio rerio*, red), in response to extracellular acidification (pH 5.5) and 10–160 mM $NH_4^+$ (neutral pH), with 1 mM $Zn^{2+}$ applied as indicated. **b** Average data (mean ± SEM) and scatterplot of the current magnitude at −80 mV from experiments as in **a**. $n = 3$ cells for MmOTOP1, $n = 4$ cells for HsOTOP1, $n = 7$ cells for GgOTOP1, and $n = 6$ cells for DrOTOP1.

be attributed to the activation of the proton channel OTOP1 in Type III TRCs. We provide a mechanism by which $NH_4Cl$ generates inward currents through OTOP1 channels; $NH_4Cl$ alkalinizes the cell cytosol, creating a driving force for proton entry while at the same time acting through an intracellular arginine residue to increase current magnitudes, possibly by gating the channels (Fig. 7). Behavioral aversion to $NH_4Cl$ of mice with a knockout of their *Otop1* gene was significantly reduced in a wild-type background and completely abolished in a *Skn-1a⁻/⁻* background. Thus, we conclude that OTOP1 is a sensor for a component of $NH_4Cl$ taste mediated by Type III TRCs.

### Ammonium sensing across species
The ability to detect and respond to environmental ammonium/ammonia is common to animals from *C. elegans* to humans. Sensory receptors for the taste or smell of ammonium/ammonia have previously been identified in several species. For the fruit fly, *Drosophila melanogaster*, low concentrations of ammonia that signal the presence of food and are attractive are detected by olfactory neurons using an ammonium transporter, Amt[4,39,40]. Higher concentrations of ammonium are aversive to *Drosophila*, inhibiting feeding behavior through effects on bitter receptors (IRs) and other possible targets[41,42]. In mosquitos, ammonia, which is present in human skins, is attractive[43]. In *C. elegans*, ammonium/ammonia is attractive as both an olfactory and taste cue[3]. It is worth noting that some of the effects of ammonium

in insects have been attributed to the basic pH of an $NH_4OH$ solution used in these experiments, rather than a direct role of ammonium[44]. OTOP channels are also expressed in invertebrates[45] and in the fruitfly *Drosophila melanogaster* they function as proton channels to mediate behavioral avoidance of acids[14,46,47]; whether they play a role in $NH_4Cl$ sensing is not known.

Most vertebrate species find the taste of ammonium aversive, which serves, in part, to protect them from eating waste products and decaying organic material. Our results are consistent with the detection of $NH_4Cl$ by both Type III (sour) TRCs and another cell type, dependent on *Skn-1a* expression. *Skn-1a* is expressed in TRCs that mediate bitter, sweet, umami, and sodium tastes, as well as a host of other TRPM5-expressing cells including brush cells in the trachea, microvillous cells in the olfactory epithelium, and solitary chemosensory neurons in the respiratory epithelium[26,27,48], any of which could mediate the *Skn-1a* dependent component of the behavioral response to $NH_4Cl$. Only in a double knockout that eliminates *Skn-1a⁻/⁻* and inactivates the Otop1 gene do we observe a complete behavioral insensitivity to high concentrations of $NH_4Cl$ as measured in brief access taste tests. In contrast, chorda tympani nerve responses, which reflect the activity of taste receptor cells in the anterior tongue, are strongly attenuated in the single Otop1 knockout mouse. One possibility, consistent with human psychophysics, is that the *Skn-1a*-dependent component of the behavioral response is mediated by

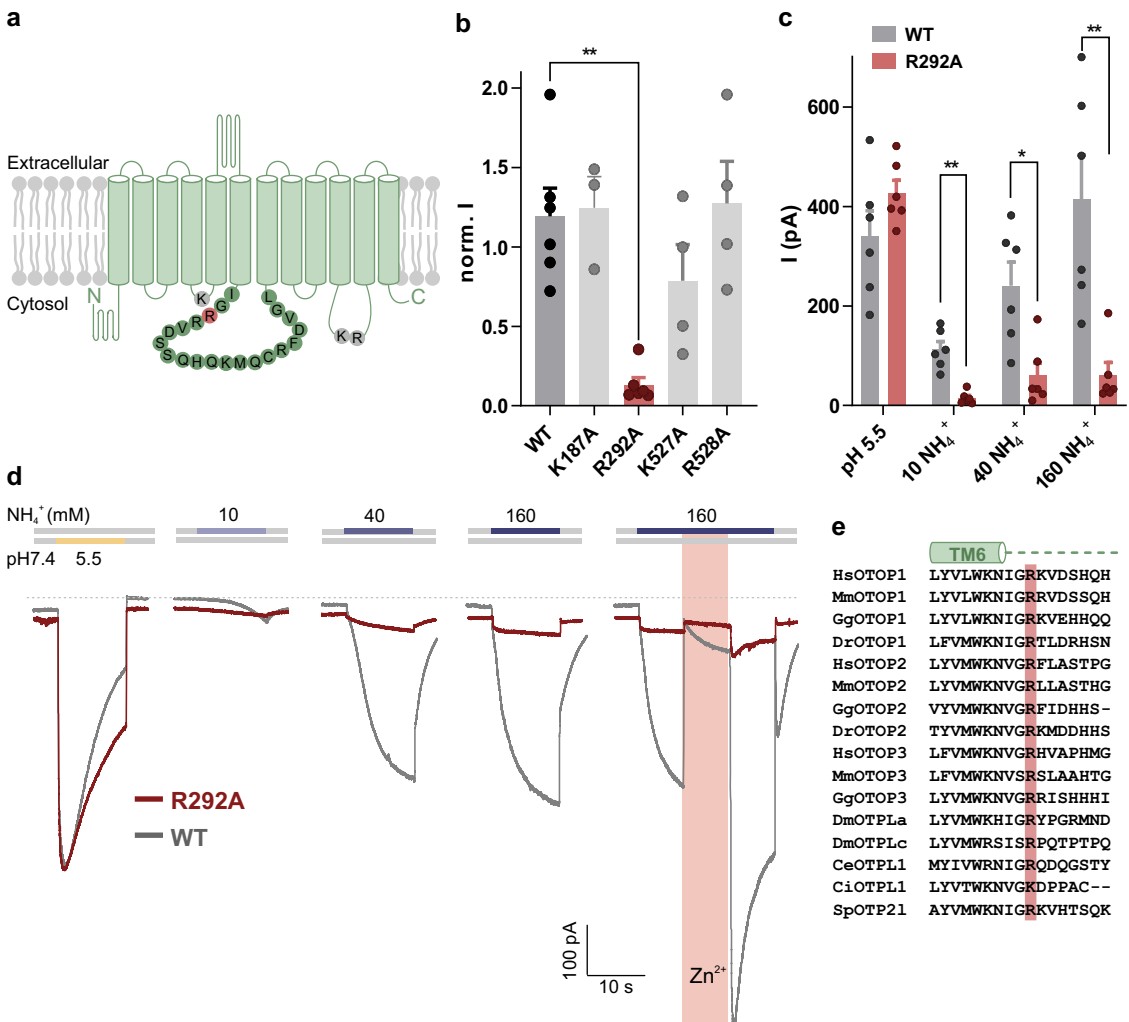

**Fig. 5 | A conserved arginine (R292) is involved in activation of mOTOP1 by NH₄Cl. a** Topology of mOTOP1 showing 12 transmembrane domains. Amino acid residues located on intracellular loops selected for mutation are highlighted in gray (K187, K527, R528) and red (R292). **b** Average data (mean ± SEM) and a scatterplot of the normalized current magnitude in response to extracellular 160 mM $NH_4^+$ from wild-type mOTOP1 and mutants (R292A, K527A, K187A, R528A). Current magnitudes were normalized to the response to the acid stimuli (pH 5.5) in each cell. Statistical significance was determined using a two-tailed Mann–Whitney test. **p < 0.01. n = 6 cells for WT and R292A, n = 4 cells for K527A and R528A, n = 3 cells

for K187A. **c** Average data (mean ± SEM) and scatterplot of the current magnitude of wild-type mOTOP1 (n = 6) and mutant R292A (n = 6) from the experiments described as in **d**. Statistical significance determined using two-tailed Welch's t test. *p < 0.05, **p < 0.01. **d** Representative trace ($V_m = -80$ mV) of HEK-293 cells expressing wild-type mOTOP1 (gray, same trace as in Fig. 4a) or mutant R292A (red) showing current in response to extracellular pH 5.5, 10–160 mM $NH_4^+$ and 1 mM $Zn^{2+}$, respectively. **e** Sequence alignment of the N-terminal portion of the tm 6–7 linker from OTOP channels from diverse species. The amino acid residues at the position equivalent to R292 in mOTOP1 are highlighted in pink.

bitter TRCs, which are more concentrated in the posterior tongue. Bitter TRCs respond to high concentrations of KCl and NaCl[49], but it remains to be determined if they respond to NH₄Cl and contribute to its detection by the gustatory system. Our observation that taste aversion to NH₄Cl is completely lost in the double knockout suggests that the trigeminal system, which can detect NH₃ gas, possibly through activation of TRPA1 or TRPV1 channels[50,51], does not redundantly detect the small amount of NH₃ available in the taste solutions and instead may only mediate responses to noxious concentrations of NH₃.

We observed that human and mouse OTOP1 channels were strongly activated by NH₄Cl, generating currents that were similar in magnitude to the currents generated in response to acids. The chicken OTOP1 channels were more sensitive and zebrafish OTOP1 channels were less sensitive to NH₄Cl. An interesting possibility is that these species differences reflect the different ecological niches and/or dietary preferences of each of these organisms. For example, birds are known to be less sensitive to sour taste, while they need to avoid ingesting NH₄Cl present in their excrement. Interestingly, naked mole

rats which live in a high ammonium environment can detect but do not avoid ammonia fumes[52], likely due to the inactivation of more general nociceptive signaling; compared with mice their taste system is less sensitive to acids but if they are also less sensitive to the taste of NH₄Cl is not known[53].

**Structural elements involved in the regulation of OTOP channels**
Our results also point to a previously undescribed mechanism through which OTOP channels are regulated by intracellular pH. A single OTOP channel subunit contains twelve transmembrane helices that are divided into two structural homologous N and C domains. The channels assemble as homodimers with the N and C domains arranged around a central lipid-filled cavity and as many as three structurally distinct permeation pathways per monomer[54,55]. Previous work has shown that some OTOP channels are gated by lowering extracellular pH, with a steep increase in currents below pH₀ = 6.0 for mOTOP1 and below pH₀ = 5.5 for mOTOP3[37]. mOTOP1 channels are also gated by raising

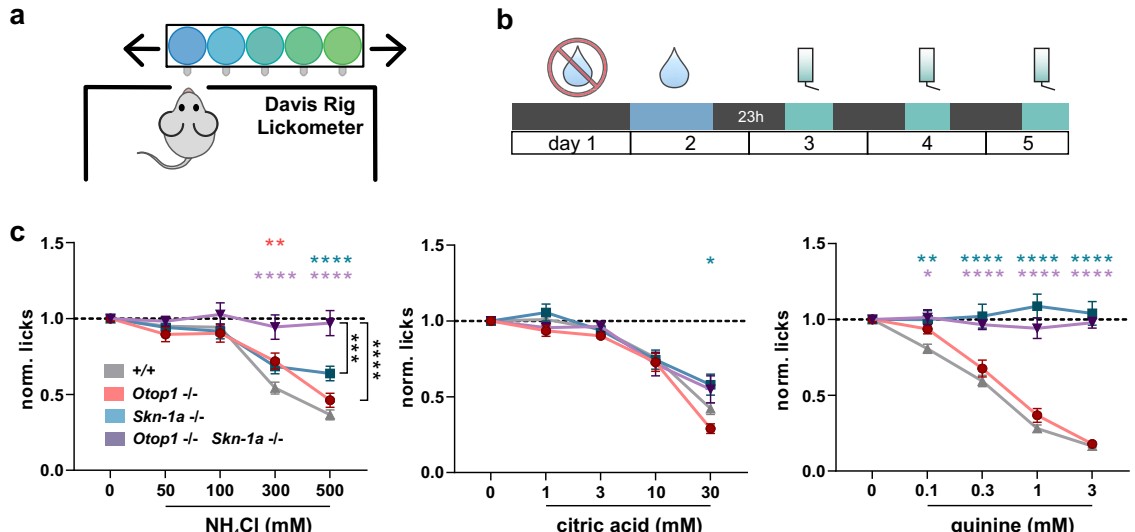

Fig. 6 | OTOP1 is necessary for the behavioral aversion to NH₄Cl mediated by Type III TRCs. a Schematic showing the mouse taste behavior setup using a Davis rig lickometer. In each experiment, mice were given the opportunity to lick at the control solution (artificial saliva), and multiple concentrations of the same chemical tastant, presented in random order. b Mice were water-deprived for 24 h (day 1), acclimatized to the rig on day 2, and then tested for three consecutive days with 23 h water deprivation between them. c Average data (mean ± SEM) of lick numbers, normalized to that of artificial saliva, for NH₄Cl (50–500 mM), citric acid (1–30 mM), and quinine (0.1–3 mM). Mouse strains were wild-type (gray), *Skn-1a*⁻/⁻ (blue), *Otop1*⁻/⁻ (red), and *Skn-1a*⁻/⁻ *Otop1*⁻/⁻ (double knockout; purple). *Otop1*⁻/⁻ mice

showed a reduction in taste aversion to 300 mM NH₄Cl; *Skn-1a*⁻/⁻ mice, were still sensitive to but showed a reduced aversion to 300 and 500 mM NH₄Cl. *Skn-1a*⁻/⁻ *Otop1*⁻/⁻ showed no aversion to any of the concentrations of NH₄Cl, while they retained aversion to citric acid. Mice lacking Type II TRCs *Skn-1a*⁻/⁻ showed no aversion to the bitter compound quinine, as expected. Statistical significance as compared with wild-type determined by two-way ANOVA with TUKEY correction for multiple comparisons. n = 36(left, right) and 37(middle) mice for wild-type, n = 15 (left, right) and 16 (middle) mice for *Otop1*⁻/⁻, n = 13 (left) and 12 (middle, right) mice for *Skn-1a*⁻/⁻, n = 8 (left, middle, right) mice for double knockout.

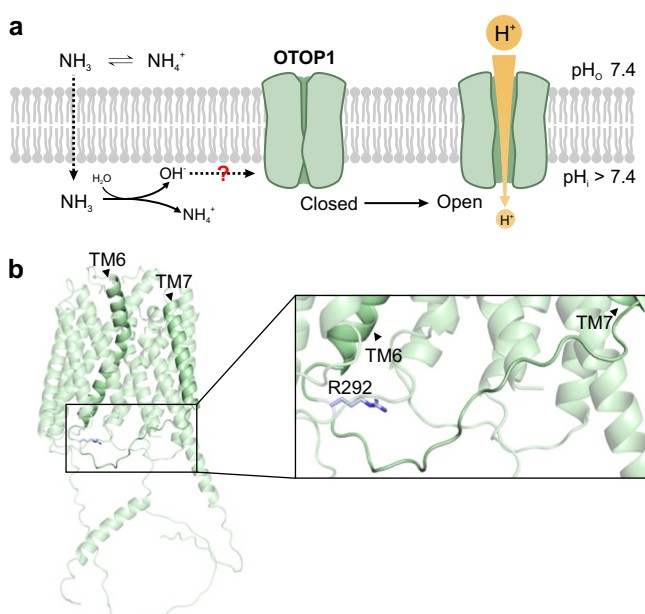

Fig. 7 | Mechanism of NH₄Cl-induced currents through OTOP1. a Proposed mechanism for activation of OTOP1 by NH₄Cl. NH₃ crosses the cell membrane inducing intracellular alkalinization. This creates both a driving force for proton entry and is hypothesized to open OTOP1 channels through actions on a conserved basic residue. b mOTOP1 structure (sideview with zoom-in) generated by Alpha-Fold2 with R292 highlighted as the potential site for intracellular pH regulation of gating of OTOP1.

extracellular pH above ~pH₀ = 8.5[37,56] and mOTOP1 and mOTOP3 channels are gated by extracellular Zn²⁺[36]. There is also some circumstantial evidence that OTOP3 channels may be closed or

inactivated when intracellular pH is lowered[55], such as might occur in response to large inward currents through the channels. Residues involved in the gating by extracellular Zn²⁺ and pH have been localized to linkers joining tm 5-6 (from the N domain) and tm 11-12 (C domain)[36,37,57]. Here we show that NH₄Cl induces large OTOP1 currents, an effect which requires an intracellular arginine residue (Fig. 7b). We show that a charge-neutralizing mutation of R292, in the tm 6-7 linker, dramatically reduces the magnitude of the NH₄Cl evoked current with little change in the magnitude or kinetics of the acid-induced currents. Extracellular NH₄Cl under the conditions of our experiments is expected to raise the intracellular concentration of NH₃/NH₄⁺ and as a consequence alkalinize the cell cytosol. It is presently unclear how changes in the intracellular milieu act on the arginine residue, or interacting partners, to increase the OTOP1 currents. It is interesting to note that this residue is positioned on the opposite side of tm 6 from residues in extracellular linkers involved in Zn²⁺ and extracellular pH regulation of OTOP1 and OTOP3[36,56,57], suggesting that they may act through a common structural rearrangement. The detailed structural basis for the activation of OTOP1 by NH₄Cl remains to be uncovered and may provide insights more generally into ion permeation and selectivity of this unique family of ion channels.

OTOP channels are expressed throughout the body and it is therefore worth considering whether they may function to either respond to or possibly transport, NH₄Cl in other cellular contexts. For example, OTOP1 was first identified in the[58] vestibular system, where it is essential for the formation of calcium carbonate-based otoconia/otoliths[59,60] and it is also expressed in brown adipose cells[14,61]. More recently an OTOP channel was shown to play a role in biomineralization in sea urchin larva[58]; it is unclear whether OTOP channels would see NH₄Cl in these cell types/tissues. On the other hand, vertebrate OTOP2 and OTOP3 have been reported to be expressed in the intestinal tract[14,62] where bacteria produce ammonia as a breakdown

product of ingested proteins. Thus, it is likely that OTOP channels mediate responses to ammonia/ammonium under other contexts in addition to taste.

## Methods

### Cell lines

HEK-293 cells (ATCC, CRL-1573) and HEK-293 carrying a mutation in the PAC1 channels (PAC-KO cells; generously provided by Dr. Zhaozhu Qiu)[63] were grown in a humidified incubator at 37 °C in 5% CO2 and 95% O2. The cells were cultured in high glucose DMEM (ThermoFisher, 11995073) containing 10% fetal bovine serum (Life Technology, 16000044), and 50 mg/mL gentamicin (Life Technology, 15750060). Cells were passaged every 3–4 days, at 1:5 or 1:10 dilutions. HEK-293 cells are tested by the manufacturer for mycoplasma contamination and subjected to STR Profiling for authentication.

### Animal strains

All animal procedures were approved by the Institutional Animal Care and Use Committees of either the University of Southern California or the University of Colorado School of Medicine. The mouse strain PKD2L1-YFP is a BAC transgenic line in which the promoter of *Pkd2l1* drives the expression of YFP as previously described[33]. The Otop1-KO mouse is a deletion of 38 bp from the 5' end of the Otop1 gene and was previously described[16]. *Skn-1a*[−/−] mice were generously provided by Ichiro Matsumoto (Monel Chemical Senses Center)[26]. Mice used in all experiments comprised both males and females. For nerve recording and behavior, mice were from the following genotypes: *Otop1*[−/−], *Skn-1a*[−/−], *Otop1*[−/−] x *Skn-1a*[−/−], wild-type littermates and compatible background wild-type mice, ages between 6 and 30 weeks. Mice were housed at the University of Colorado Anschutz Medical Campus on a 12 hr light/dark cycle and had continual access to standard chow. Temperature was maintained at 72 °C and >30% humidity. For isolated taste receptor cells, mice were from the following genotype: *Otop1*[−/−] and *Otop*[+/+] in a PKD2L1-YFP background for cell identification. Mice were housed at the University of Southern of California on a 12 hr light/dark cycle and had continual access to standard chow. Temperature was maintained at 68 °C–74 °C and humidity was 30–70%.

### Taste cell isolation

Taste cells were isolated from adult mice (6–12 weeks) as previously described[12,33]. Briefly, 0.3 mL enzyme mix consisting of Tyrode's solution supplemented with 1 mg/mL elastase (Worthington Biochemical, Cat# LS002290), and 2.5 mg/mL Dispase II (Sigma Aldrich, Cat# D4693), was injected between the epithelium and muscle of isolated tongue, which was then incubated in Tyrode's solution for 20 min at room temperature. The epithelium was then peeled off from the tongue. A small piece of the epithelium containing the circumvallate papillae was minced and incubated in the enzyme mix for another 15 min at room temperature.

The mixture containing circumvallate papillae was centrifuged for 2 min and the supernatant was removed. The resulting precipitate was resuspended with $Ca^{2+}$-free Tyrode's solution (145 mM NaCl, 10 mM HEPES, 0.5 mM EGTA). The mixture was then centrifuged again for 2 min and the supernatant was replaced with Tyrode's solution. Single TRCs were further isolated by trituration in Tyrode's solution with fire-polished Pasteur pipettes and were used in the following 4–6 h.

### Transfection of HEK-293 cells and PAC-KO cells, clones, and constructs

HEK-293 (PAC-KO) cells were transfected in 35 mm Petri dishes, with ~600 ng DNA and 2 μL TransIT-LT1 transfection reagent (Mirus Bio Corporation, Cat# MIR2300) following the manufacturer's protocol. The cells were lifted using Trypsin-EDTA 20-28 h after transfection and plated onto a coverslip for patch-clamp recordings.

For Fig. 3a, f, cDNA encoding mOTOP1 was co-transfected with eGFP 5:1 to allow for the selection of transfected cells. For Fig. 3d, e, the same construct was co-transfected with pHlourin (1:1). For Fig. 3, mOTOP1 cDNA was in pcDNA3.1[14]. For Figs. 4 and 5, and Supplementary Fig. 1, cDNAs were subcloned into a pcDNA3.1 vector that adds an N-terminal fusion tag consisting of an octahistidine tag followed by eGFP, a Gly-Thr-Gly-Thr linker and 3 C protease cleavage site (LEVLFQGP)[54]. Zebrafish *Otop1* was as previously described[54], Chicken *Otop1* was synthesized by Twist biosciences, codon-optimized, based on the protein accession number (XP_025005572) and human and mouse cDNAs were as descsribed[14]. Single point mutations were introduced using In-Fusion Cloning (Takara) using primers generated by IDT (see Supplementary Table 1) and sequences were confirmed by Sanger sequencing (Genewiz).

### Patch-clamp electrophysiology

Whole-cell patch-clamp recording was employed following the protocol previously described by[14,33]. The recordings were performed with an Axonpatch 200B amplifier, digitized with a Digidata 1322a 16-bit data acquisition system, acquired with pClamp 8.2, and analyzed with Clampfit 8.2 (Molecular devices). A sampling rate of 5 kHz and a 1 kHz filter were applied during the recordings. Patch pipettes with a resistance of 2–5 MOhm were fabricated from borosilicate glass and only recordings with a giga-Ohm seal were used in the analysis. The membrane potential was maintained at −80 mV for most of the experiments. Cells were lifted and placed in front of an array of microcapillary tubes controlled by a Fast-Step perfusion system (Warner Instruments) through which the extracellular solution could be rapidly exchanged (flow rate of 40 μL/min).

Cell-attached recordings on TRCs were performed as previously described[33]. Type III TRCs exhibiting a characteristic flask-shaped morphology, intact apical processes, and YFP expression were selectively subjected to cell-attached recordings. The number of action potentials was measured during the initial 2 seconds of stimulus application, unless stated otherwise, to quantify the results. Inhibition by $Zn^{2+}$ was measured as: 1-(APs during $Zn^{2+}$)/(APs before $Zn^{2+}$). As a control, the overall electrical excitability of TRCs was evaluated by exposure to a high $K^+$ extracellular solution, a Tyrode's-based solution in which 20 mM $Na^+$ was substituted with $K^+$ for a final concentration of 25 mM $K^+$. Cells that did respond to high $K^+$ were excluded from further analysis. Solutions were delivered by a Fast-Step perfusion apparatus (Warner Instruments).

To calculate $pH_i$ in experiments in which we measured $E_{rev}$ (Fig. 3f), we used the Nernst equation: We assume $E_{rev} = E_{H+}$ so that $E_{rev} = 59 \log ([H^+]_o/[H^+]_i) = 59(pH_o - pH_i)$. For $pH_o = 7.4$, $pH_i = 7.4 + (E_{rev}/59)$.

### Patch-clamp electrophysiology solutions

For experiments in Fig. 1, standard Tyrode's solution consisted of 145 mM NaCl, 5 mM KCl, 1 mM $MgCl_2$, 2 mM $CaCl_2$, 20 mM D-Glucose, and either 10 mM HEPES (for pH 7.4) or 10 mM MES (for pH 6.5). To evaluate the excitability in control experiments, 20 mM $Na^+$ in Tyrode's solution was substituted with an equimolar concentration of $K^+$ resulting in a final KCl concentration of 25 mM. Modified Tyrode's solution buffered at pH 6.5 consisted of 145 mM NaCl, 5 mM KCl, 1 mM $MgCl_2$, 2 mM $CaCl_2$, 20 mM D-Glucose, and 10 mM MES. When measuring the response to extracellular $NH_4Cl$ in cell-attached AP recordings, equimolar concentrations of $Na^+$ were substituted with 5–20 mM $NH_4^+$ as indicated in the figures.

For experiments in Fig. 2, the isolated TRCs were bathed in a standard Tyrode's solution at pH 7.4 and APs were evoked by external stimulations of pH 6.5 Tyrode's solution and 10 mM $NH_4Cl$ solution as described in Fig. 1 respectively. To test the $Zn^{2+}$ inhibition of APs evoked by extracellular acidification or $NH_4Cl$, 0.1 μM-1 mM $ZnCl_2$ were added into pH 6.5 Tyrode's solution and 10 mM $NH_4Cl$ solution, respectively. The external solutions containing $ZnCl_2$ were buffered to

their original pH. For Figs. 1 and 2, the pipette solution for cell-attached recording contained the standard Tyrode's solution at pH 7.4 as described in Fig. 1.

For the whole-cell patch-clamp experiments in Figs. 3–5 and Supplementary Fig. 1, standard NMDG-based extracellular solutions contained 160 mM NMDG-Cl, 2 mM $CaCl_2$, and either 10 mM HEPES (for pH 7.4) or 10 mM MES (for pH 6–5.5), pH adjusted with HCl. When measuring the response to extracellular $NH_4Cl$ in whole-cell recording, equimolar concentrations of $NMDG^+$ were substituted with 1–160 mM $NH_4^+$. To test the $Zn^{2+}$ inhibition of $NH_4Cl$ evoked OTOP1 current, 0.3 μM-1 mM $ZnCl_2$ was added into 160 mM $NH_4Cl$ solution and buffered to pH 7.4. The pipette contained a standard internal solution consisting of 120 mM Cs-aspartate, 15 mM CsCl, 2 mM Mg-ATP, 5 mM EGTA, 2.4 mM $CaCl_2$ (100 nM free $Ca^{2+}$), and 10 mM HEPES with a pH of 7.3 adjusted using CsOH and an osmolarity of 290 mOsm. All external solutions used were adjusted to an osmolarity of 300 mOsm.

### Combined patch-clamp and pH imaging of HEK-293 cells
HEK-293 cells were cultured in 35 mm Petri dishes and co-transfected with cDNA encoding mOTOP1 and the pH-sensitive indicator pHluorin. The cells were lifted and plated on poly-D-lysine-coated coverslips after 24 h transfection. To simultaneously record the response of the pH indicator while measuring currents, once whole-cell recording mode was achieved, the cells were lifted in front of an array of microcapillary tubes (Warner Instruments) and imaging was initiated. Cells were illuminated at 488 nM and emission at 510 nm was detected using a U-MNIBA2 GFP filter cube (Olympus). Images were captured at a frame rate of 1/sec using a Hamamatsu digital CCD camera attached to an Olympus IX71 microscope and analyzed using Simple PCI software. The fluorescence intensity of each cell was measured following subtraction of the background fluorescence.

### Chorda tympani nerve recordings
To record from the chorda tympani nerve, mice were first anesthetized via an IP injection of urethane at 2 g/kg (Sigma, U2500). They were stabilized under a dissection microscope using a custom head holder, and a tracheotomy was performed to facilitate breathing during tongue stimulation. The Chorda Tympani nerve was approached ventrally, severed near the tympanic bulla, and placed on a platinum-iridium wire electrode. A reference electrode was placed into nearby muscle tissue. A motorized pump (mini-pump, variable flow; Fisher Scientific) allowed for the stimulation of the anterior tongue with a continuous flow of taste stimuli or artificial saliva. All tastants were presented in a background of artificial saliva as used in[38] and consisted of the following, pH to between 7.4 and 7.6: 4 mM NaCl, 10 mM KCl, 6 mM $KHCO_3$, 6 mM $NaHCO_3$, 0.5 mM $CaCl_2$, 0.5 mM $MgCl_2$, 0.24 mM $K_2HPO_4$, 0.24 mM $KH_2PO_4$. Artificial saliva was also perfused across the tongue during rinse/rest periods. Tastants included 100 mM $NH_4Cl$, 10 mM citric acid, 8 mM Ace K, and 500 mM KCl. Stimuli were applied for 20 seconds, interspersed between 40-second periods of rinse/rest. For Zinc block experiments, $ZnCl_2$ at 1, 3, or 10 mM was perfused onto the tongue ~20 seconds prior to and during the 20 s tastant stimulation. Nerve responses were amplified (P511, Grass Instruments), integrated over a time constant of 0.5 s, and recorded via Acknowledge software (Biopac). Responses were quantified by measuring the mean of the integrated response over 20 seconds from the onset of the stimulus and subtracting the mean of a 5 s period of rinse/rest baseline activity. All responses were then normalized to baseline nerve activity during the recording, as previously described[16].

### Taste preference
Before testing periods, mice were trained in a Davis Rig lickometer. At the start of each 5-day behavioral trial, mice were water-deprived for 23 hours before the onset of any trial. For early training, mice were given access to water in the Davis Rig without the sipper tube

door opening or closing. Once they learned to obtain their water from this source, the experimental period of 5 s sipper tube access was introduced. Once mice mastered this paradigm with water, trial weeks ensued. For each tastant group, mice were deprived of water for 23 hours before a single day of acclimatization to the apparatus with water. Mice were deprived of water again between this and all test days. For the following three test days, several concentrations of the tastant, as well as one tube of artificial saliva, were presented in random order for 5 s from the onset of mouse licking. Each mouse was permitted to continue these trials through 5 randomized full blocks of tastants before the test day concluded. A tastant block consisted of all presented tastants, and during each block, the tastants were presented in a random order. Lick numbers for each tastant in each block were tabulated and the average across all blocks was normalized to the average licks of artificial saliva. Only full (complete) blocks were used. Data was excluded if the average licks to artificial saliva on a day was <20, which represents <50% of the maximal lick rate. Tastant concentrations were as follows: for $NH_4Cl$−50, 100, 300, and 500 mM; for citric acid−1, 3, 10, and 30 mM; for quinine (bitter)−0.1, 0.3, 1, and 3 mM. All tastants were presented in a background of artificial saliva.

### Quantification, statistical analysis and figure making
All data are presented as mean ± SEM unless otherwise noted and each data point represents an independent biological replicant (e.g. cell or animal) unless otherwise stated. Statistical analyses (ANOVA or Student's t-test) were performed using Graphpad Prism 9.5.1 (Graphpad Software Inc) and all statistical tests were two-sided unless otherwise stated. Sample sizes in the present study are similar to those reported in the literature for similar studies. Representative data shown in the figures acquired with PClamp was in some cases decimated 10-fold before exporting into the graphics programs Origin 6.1 (Microcal) and Coreldraw 2019 (Corel). Channel structure was generated in Pymol 2.5.2 (Schrodinger Inc).

### Reporting summary
Further information on research design is available in the Nature Portfolio Reporting Summary linked to this article.

## Data availability
All data supporting the findings of this study are available within the paper and its Supplementary Information. The source data underlying Figs. 1b, c, e, 2b, d, f, 3b, c, e, f, 4b, 5b, c, 6c, and Supplementary Fig. 1b are provided as a Source Data file. The PDB structure of mOTOP1 model can be found at https://alphafold.ebi.ac.uk/entry/Q80VM9. Source data are provided with this paper.

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

## Acknowledgements
We thank Anne Tran, Kevin Chyung, and Joshua Kaplan for expert technical support and all members of the Liman and Kinnamon labs for helpful discussions and feedback on the manuscript. Supported by grants from the National Institutes of Health R01GM131234 to E.R.L. and R01DC013741 to E.R.L. and S.C.K.

## Author contributions
Z.L. designed the study, carried out electrophysiological experiments on isolated taste cells and HEK-293 cells, generated constructs, analyzed the data, and wrote the paper, C.E.W. designed the study, carried out gustatory nerve recordings and behavioral experiments, analyzed the data, and wrote the paper, B.T. designed the study, carried out electrophysiological experiments on HEK-293 cells and analyzed the data, S.C.K and E.R.L designed the study, analyzed the data and wrote the paper.

## Competing interests
The authors declare no competing interests.
