## [Peer Review File · Nature Communications]

The Proton Channel OTOP1 is a Sensor for the Taste of Ammonium ChlorideReviewers' Comments:

Reviewer #1:

Remarks to the Author:

This is an important manuscript that describes the role of the recently identified proton-selective ion channel, OTOP1, in the sensation of ammonium in murine taste buds. The authors employ multidisciplinary approaches that span channel biophysics, cellular and neural physiology, and psychophysics, involving multiple transgenic animal models and pharmacological interventions. The methodology employed is well thought-out and executed to a high standard. They demonstrate requirement of OTOP1 channels in the chorda tympani nerve and type III taste cell responses to ammonium chloride (NH₄Cl) using Otop1 KO mice and Zn²⁺, a blocker of OTOP1 channel. Using a heterologous expression system and combining patch clamp electrophysiology and intracellular pH imaging, they further demonstrate that NH₄Cl-induced intracellular alkalinization creates a driving force for proton entry through OTOP1 channels, thereby depolarizing the cells. Moreover, the authors identify a critical intracellular arginine residue of mouse OTOP1 that enables the channel to sense intracellular alkalinization and regulate its gating. They also find that this ammonium response, along with the arginine residue, is conserved across OTOP1 channels in multiple vertebrate animals. Finally, the manuscript provides compelling evidence for the involvement of OTOP1 in the behavioral aversion of NH₄Cl, as demonstrated through brief access lickometer tests. Their results unambiguously reveal the ammonium sensing mechanism in type III taste cells with OTOP1 serving as the primary sensor, while also suggesting the presence of as-yet-unknown sensors in type II taste cells.

The manuscript is well-written and the method section provide sufficient details to facilitate reproducibility of the results by other researchers.

In the taste field, ammonium sensing has been known and used for decades, but its underlying mechanism was unknown. This identification of a fundamental mechanism for animals to detect environmental chemicals advances the field and is of great interest to a broad readership of Nature Communications.

This is an impressive piece of work, and I have only minor comments and suggestions:

1. Lines 63-64: The classification of taste cells into type I/II/III is primarily based on morphology. While there are other features that suggest such categorization (PMCID: PMC7729297), it is not currently accurate to categorize that TRCs mediating sodium taste as type II cells, as the morphological characteristics of sodium cells are still unknown.
2. Line 139: Does the slow-rising phase observed in the OTOP1 current in Fig.3 represent the slow channel gating as tested in Fig.5? I am confused as to whether the authors still lack an explanation or have one now. If this phase represents gating, it would be beneficial to analyze the relationship between the kinetics of OTOP1 current and the stimulus intensity, which would provide further mechanistic insights.
3. In the whole cell patch clamp recordings, the intracellular solution contains 10 mM HEPES. Could the pH buffer potentially have influenced the extent of intracellular alkalinization, particularly when low concentrations (1-5 mM) of NH₄Cl were applied?
4. Throughout the manuscript: Please use "Skn-1a" or "Skn1a" consistently throughout the text.
5. Line 63: one "that" should be removed
6. Line 99: Otop1-/- -> Otop1-/-
7. Line 105: taste receptor cells -> TRCs
8. Lines 105-110: It would be helpful to mention the IC₅₀ value of Zn on the OTOP1 channel to facilitate a comparison of Zn effects between channels, isolated TRCs, and the tongue.
9. Lines 110-112: Please provide a reference regarding this dilution factor, if possible.
10. Line 123: NHeCl -> NH₄Cl
11. Line 357: Is the final K⁺ concentration 30 mM? While 25 mM Na⁺ was substituted with an

equimolar concentration of K⁺, the original Tyrode's composition included 5 mM K⁺.

12. Line 371: 1-20 -> 5-20? Based on Fig. 1d.

13. Line 418: Something is missing in the following sentence "Only full (complete) were used"

14. Fig. 1d: It would be helpful if the unit of NH₄⁺ is indicated in the figure or legend

15. Fig. 3f: What is the middle small gray trace? Is it a voltage trace or a current trace from untransfected cells? Please provide clarification.

16. Fig.3f: Please provide details for the calculation of pHi in the method section or legend

17. Fig. 4a: time scale is missing

18. Fig. 6c: Sn1a -> Skn1a

19. Statistical test: In Fig.1bc and Fig.6c, please specify post-hoc tests employed.

20. Method: Please specify the source of OTOPI sequences from these species.

21. Method (Solutions): I assumed "standard NMDG-based extracellular solutions" were used in the whole cell patch clamp experiments in HEK cells. Please specify clearly which solution was used in each experiment. Also, the recipe for pH6.5 extracellular solution is not provided.

Reviewer #2:

Remarks to the Author:

The work by Liang et al., represents a novel and important contribution to the field of sensory neuroscience. Their major finding is that ammonium, an ecologically relevant molecule that many animals can sense with their taste systems, is detected in part through otopetrin, a proton channel best known for its role in acid-sensing type III taste receptor cells. Using a series of electrophysiology experiments in taste receptor cells and heterologous cells, the authors demonstrate that NH₄Cl increases H⁺ currents through otopetrin due to the ability of ammonia gas (NH₃) to cross the plasma membrane and alkalinize the cell. The alkalization induces H⁺ currents through otopetrin through two means. First, the alkalization increases the H⁺ concentration gradient for protons to enter the cell. Additionally, the alkalization affects a cytosolic arginine residue on otopetrin, gating the channel that is mostly closed normally. At the whole animal level, they demonstrate that ammonium responses in a nerve innervating the taste buds is mostly lost in otopetrin knockouts. However, behavioral taste responses to NH₄Cl are mostly unaffected in the otopetrin knockout, due to redundancy with other ammonium sensing mechanisms, likely mediated by type II taste receptor cells.

Concerns

1. The gating model is reasonable, but additional evidence is required. For example, a slight lowering of pH (to 6.5 or 7) that doesn't normally lead to a current through otopetrins, should lead to a large increase in H⁺ current in the presence of ammonium chloride due to increased channel gating and greater H⁺ concentration gradient. The effects of NH₄Cl on mild acid responses should be greatly reduced in the R292A otopetrin mutant.

If possible, the authors should alkalinize the cytosolic pH independent of ammonium and measure currents through otopetrin in order to determine whether otopetrin responds to intracellular basic pH rather than ammonium directly, as their model suggests.

2. Given the mechanisms proposed, the title of the article is misleading, as otopetrin is not an ammonium receptor per se (doesn't directly bind or interact with ammonium), but is rather a sensor of alkalization induced by ammonium. Similar wording is found in line 5 of the abstract, 118 and 163 of the text, and other locations. A more accurate title such as "The proton channel OTOPI is essential for the response of type III taste receptor cells to ammonium" or "The proton channel OTOPI contributes to the taste response to ammonium" should be used.

3. The behavioral data clearly indicate that NH₄Cl is redundantly sensed by both type II and type III TRCs since the response is only slightly dulled in otopetrin KOs or Skn-1a KOs (lacking type II cells), but abolished in the double KO. The authors suggest this is due to bitter TRCs, which are not well represented by their chorda tympani recordings responsive to taste buds in the front of the tongue. Ideally the authors could record from bitter taste receptor cells to substantiate this interpretation since

skn-1a is a transcription factor expressed in a wide number of chemosensory cells outside tongue (see for example <https://www.annualreviews.org/doi/abs/10.1146/annurev-immunol-042718-041505> and <https://pubmed.ncbi.nlm.nih.gov/29216297/>). Thus, behavioral responses could alternatively be mediated by such cells instead of bitter TRCs.

If the authors do not directly show that type II TRCs are involved, these alternative possibilities should be considered in their discussion.

4. The abstract and introduction suggest that otopetrin is the primary means by which ammonium chloride is detected by the taste system. However, as noted above, their behavioral data suggest otherwise. The abstract and introduction should be re-written to more clearly indicate that otopetrin is one mechanism by which ammonium chloride is detected by the taste system, but that there are others.

Response to reviewers.

We thank the reviewers for their careful review of our manuscript. We are pleased that they found the work “important,” and “of great interest to a broad readership of Nature Communications.” We have addressed all the reviewers’ comments and concerns as described below with edits throughout the text and figures and with one additional experiment (Supplementary Figure 1).

REVIEWER COMMENTS

Reviewer #1 (Remarks to the Author):

This is an important manuscript that describes the role of the recently identified proton-selective ion channel, OTOP1, in the sensation of ammonium in murine taste buds. The authors employ multidisciplinary approaches that span channel biophysics, cellular and neural physiology, and psychophysics, involving multiple transgenic animal models and pharmacological interventions. The methodology employed is well thought-out and executed to a high standard. They demonstrate requirement of OTOP1 channels in the chorda tympani nerve and type III taste cell responses to ammonium chloride (NH₄Cl) using Otop1 KO mice and Zn²⁺, a blocker of OTOP1 channel. Using a heterologous expression system and combining patch clamp electrophysiology and intracellular pH imaging, they further demonstrate that NH₄Cl-induced intracellular alkalinization creates a driving force for proton entry through OTOP1 channels, thereby depolarizing the cells. Moreover, the authors identify a critical intracellular arginine residue of mouse OTOP1 that enables the channel to sense intracellular alkalinization and regulate its gating. They also find that this ammonium response, along with the arginine residue, is conserved across OTOP1 channels in multiple vertebrate animals. Finally, the manuscript provides compelling evidence for the involvement of OTOP1 in the behavioral aversion of NH₄Cl, as demonstrated through brief access lickometer tests. Their results unambiguously reveal the ammonium sensing mechanism in type III taste cells with OTOP1 serving as the primary sensor, while also suggesting the presence of as-yet-unknown sensors in type II taste cells.

The manuscript is well-written and the method section provide sufficient details to facilitate reproducibility of the results by other researchers.

In the taste field, ammonium sensing has been known and used for decades, but its underlying mechanism was unknown. This identification of a fundamental mechanism for animals to detect environmental chemicals advances the field and is of great interest to a broad readership of Nature Communications.

This is an impressive piece of work, and I have only minor comments and suggestions:

1. Lines 63-64: The classification of taste cells into type I/II/III is primarily based on morphology. While there are other features that suggest such categorization (PMCID: PMC7729297), it is not currently

accurate to categorize that TRCs mediating sodium taste as type II cells, as the morphological characteristics of sodium cells are still unknown.

Response: Corrected. The reviewer is correct that the sodium (salty) taste cells have not been associated with one of the morphologically defined classes of taste cells.

2. Line 139: Does the slow-rising phase observed in the OTOP1 current in Fig.3 represent the slow channel gating as tested in Fig.5? I am confused as to whether the authors still lack an explanation or have one now. If this phase represents gating, it would be beneficial to analyze the relationship between the kinetics of OTOP1 current and the stimulus intensity, which would provide further mechanistic insights.

Response: Line 139 was removed. The reviewer is correct that the slow phase of the currents may reflect the gating of the channels as it is not observed in the mutant as we describe. We did some preliminary analysis of the kinetics of the responses which shows that the slow component becomes a smaller fraction of the total as the stimulus intensity increases, as would be expected if channels required a more mild alkalinization to be fully gated (maximally activated) and additional alkalinization increases driving force. However, there are many free parameters, and the fits are not perfect so that we do not feel comfortable including this data in the present manuscript. See response to Rev 2 for further discussion of this and related issues.

3. In the whole cell patch clamp recordings, the intracellular solution contains 10 mM HEPES. Could the pH buffer potentially have influenced the extent of intracellular alkalinization, particularly when low concentrations (1-5 mM) of NH₄Cl were applied?

Response: The main point of Figure 1d is to show that the activation of the OTOP1 correlates roughly with the degree of intracellular alkalinization. We do not mean to imply that these specific measurements would apply to an intact cell, which will have different buffers, although from a few experiments (not shown) where we have varied the intracellular buffer concentration, we don't think the responses would be very different. Indeed, the responses in the HEK cells (dialyzed with the HEPES buffered intracellular solution) which show an ~ doubling of pH response and currents from the 5 mM to 20 mM NH₄Cl correlates well to the change in AP firing observed in the taste cells (that are not dialyzed) over the same concentration range (also a doubling).

4. Throughout the manuscript: Please use "Skn-1a" or "Skn1a" consistently throughout the text.

Response: fixed

5. Line 63: one "that" should be removed

Response: fixed

6. Line 99: Otop1-/-  Otop1-/-

Response: fixed

7. Line 105: taste receptor cells -> TRCs

Response: fixed

8. Lines 105-110: It would be helpful to mention the IC50 value of Zn on the OTO1 channel to facilitate a comparison of Zn effects between channels, isolated TRCs, and the tongue.

Response: This is an excellent point. Previous data regarding IC50 values for Zn inhibition of OTO1 currents in taste cells and HEK293 cells expressing OTO1 were obtained under conditions of acidic pH, which lowers Zn affinity (Bushman et al, 2016). To be able to compare data between OTO1 currents and taste cells, we have now measured Zn inhibition of NH4Cl-evoked OTO1 currents. The data, now shown in supplementary fig 1 gives an IC50 of 1.3 μ M, very similar to the IC50 for inhibition of APs in taste cells of \sim 1 μ M.

9. Lines 110-112: Please provide a reference regarding this dilution factor, if possible.

Response: changed wording so it is clear this was not a known fact, but comes from our data.

10. Line 123: NHeCl -> NH4Cl

Response: fixed

11. Line 357: Is the final K+ concentration 30 mM? While 25 mM Na+ was substituted with an equimolar concentration of K+, the original Tyrode's composition included 5 mM K+.

Response: The total K concentration is 25 mM. 20 mM Na+ is substituted in Tyrode's with equimolar K+. This is now described.

12. Line 371: 1-20 -> 5-20? Based on Fig. 1d.

Response: fixed

13. Line 418: Something is missing in the following sentence "Only full (complete) were used"

Response: fixed – "blocks" was missing

14. Fig. 1d: It would be helpful if the unit of NH4+ is indicated in the figure or legend

Response: fixed – unit is given in the figure legend

15. Fig. 3f: What is the middle small gray trace? Is it a voltage trace or a current trace from untransfected cells? Please provide clarification.

Response: Thank you for pointing out that this needed to be labeled. It is now corrected, and labeled to show that it is the voltage protocol.

16. Fig.3f: Please provide details for the calculation of pHi in the method section or legend

Response: Now added to methods (line ~405): To calculate pHi in experiments in which we measured E_{rev} (Figure 3f), we used the Nernst equation: We assume $E_{rev} = E_{H^+}$, so that $E_{rev} = 59 \log ([H^+]_o/[H^+]_i) = 59(pH_o - pH_i)$. For $pH_o = 7.4$, $pH_i = 7.4 + (E_{rev}/59)$.

17. Fig. 4a: time scale is missing

Response: fixed

18. Fig. 6c: Sn1a -> Skn1a

Response: fixed

19. Statistical test: In Fig.1bc and Fig.6c, please specify post-hoc tests employed.

Response: P values for multiple comparisons of gustatory nerve response were adjusted by Bonferonni correction (Fig 1b,c). Similar results were obtained with Dunnett's test. Note that for Fig 1b we inadvertently showed the uncorrected P value for the comparisons, so that the significance of AceK has now been adjusted from $p < 0.01$ to $p < 0.05$.

For behavioral analyses, we used Tukey's method for multiple comparisons (recommended by Graphpad). In the process, we also redid the analysis to remove trials where the average number of licks to the control stimulus used for normalization was less than 20 (compared with a maximum of ~40). The analysis was now done completely blind to genotype, by two investigators (ZL and EL) blinded to genotype. This did not result in any change in the results (graphs look similar), but the data is now more consistent, and the statistical analysis is more reliable. Note that we could have chosen a different method for multiple comparisons, and the overall trends would be the same, although the p values would be slightly different. Tukey's is intermediate among methods in terms of stringency. This information is now added to the methods section and the data is now provided in an attachment to allow re-analyses by other researchers if so inclined.

20. Method: Please specify the source of OTOP1 sequences from these species.

Response: This information has now been added to the methods section.

21. Method (Solutions): I assumed "standard NMDG-based extracellular solutions" were used in the whole cell patch clamp experiments in HEK cells. Please specify clearly which solution was used in each experiment. Also, the recipe for pH6.5 extracellular solution is not provided.

Response: This information has now been added to the methods section.

Reviewer #2 (Remarks to the Author):

The work by Liang et al., represents a novel and important contribution to the field of sensory neuroscience. Their major finding is that ammonium, an ecologically relevant molecule that many animals can sense with their taste systems, is detected in part through otopetrin, a proton channel best known for its role in acid-sensing type III taste receptor cells. Using a series of electrophysiology experiments in taste receptor cells and heterologous cells, the authors demonstrate that NH₄Cl increases H⁺ currents through otopetrin due to the ability of ammonia gas (NH₃) to cross the plasma membrane and alkalize the cell. The alkalization induces H⁺ currents through otopetrin through two means. First, the alkalization increases the H⁺ concentration gradient for protons to enter the cell. Additionally, the alkalization affects a cytosolic arginine residue on otopetrin, gating the channel that is mostly closed normally. At the whole animal level, they demonstrate that ammonium responses in a nerve innervating the taste buds is mostly lost in otopetrin knockouts. However, behavioral taste responses to NH₄Cl are mostly unaffected in the otopetrin knockout, due to redundancy with other ammonium sensing mechanisms, likely mediated by type II taste receptor cells.

Concerns

1. The gating model is reasonable, but additional evidence is required. For example, a slight lowering of pH (to 6.5 or 7) that doesn't normally lead to a current through otopetrins, should lead to a large increase in H⁺ current in the presence of ammonium chloride due to increased channel gating and greater H⁺ concentration gradient. The effects of NH₄Cl on mild acid responses should be greatly reduced in the R292A otopetrin mutant.

If possible, the authors should alkalize the cytosolic pH independent of ammonium and measure currents through otopetrin in order to determine whether otopetrin responds to intracellular basic pH rather than ammonium directly, as their model suggests.

Response: The reviewer brings up an important issue regarding the mechanism by which NH₄Cl activates large OTO1 currents. In our initial submission, we came out strongly in favor a mechanism by which intracellular alkalization not only increases the driving force for proton entry but also gates the channels. The evidence for gating was (1) the large magnitude of the responses to NH₄Cl, (2) species differences in responses to NH₄Cl, and (3) the effect of mutating an intracellular residue on responses to NH₄Cl and not to acids. We agree that more experiments should be done, such as varying intracellular pH or extracellular pH. However, initial attempts at these experiments have revealed complexities that cloud the interpretation of any such data. For example, changing extracellular pH does not just change the pH gradient (driving force) for protons, it also increases the concentration of NH₃ (which then changes intracellular pH). Changing/controlling intracellular pH is also not straightforward, and we have only succeeded in doing so when we hold V_m close to E_H, which introduces other confounds when looking at gating (e.g., we have to rule out an effect of voltage on gating under the conditions of the experiment). This will require a much more detailed study of the channel biophysics, which is beyond the scope of the present manuscript and will ultimately need to be substantiated with structures of the channel in various states, something we are working towards.

Given that the reviewer's concerns, we have tempered our assertion that the channels are gated by intracellular alkalization, with edits throughout the text and in the model (e.g. Abstract; Section beginning around line 167; the discussion section beginning around line 265).

This does not affect our main conclusion that sensitivity to ammonium is conferred by the intracellular residues that we identified, is separable from acid sensitivity, and varies across species.

2. Given the mechanisms proposed, the title of the article is misleading, as otopetrin is not an ammonium receptor per se (doesn't directly bind or interact with ammonium), but is rather a sensor of alkalization induced by ammonium. Similar wording is found in line 5 of the abstract, 118 and 163 of the text, and other locations. A more accurate title such as "The proton channel OTOP1 is essential for the response of type III taste receptor cells to ammonium" or "The proton channel OTOP1 contributes to the taste response to ammonium" is should be used.

Response: We have changed the title to "The proton channel OTOP1 is a sensor for the taste of ammonium chloride" using some of the reviewer's wording. We prefer not to introduce jargon (e.g. type III taste receptor cells) which may make the paper less accessible to a wide readership, and the results go well beyond showing OTOP1 is necessary or essential, as we also show "sufficiency" (e.g. ability to confer sensitivity to NH₄Cl on a heterologous cell). Note that we do not say "the sensor" leaving open the possibility that there are other sensors.

Other changes: Line 5 changed "receptor" to "mechanism of action" Line 118, 163, 192– "sensory receptor" changed to "sensor."

3. The behavioral data clearly indicate that NH₄Cl is redundantly sensed by both type II and type III TRCs since the response is only slightly dulled in otopetrin KOs or Skn-1a KOs (lacking type II cells), but abolished in the double KO. The authors suggest this is due to bitter TRCs, which are not well represented by their chorda tympani recordings responsive to taste buds in the front of the tongue. Ideally the authors could record from bitter taste receptor cells to substantiate this interpretation since skn-1a is a transcription factor expressed in a wide number of chemosensory cells outside tongue (see for example <https://www.annualreviews.org/doi/abs/10.1146/annurev-immunol-042718-041505> and <https://pubmed.ncbi.nlm.nih.gov/29216297/>). Thus, behavioral responses could alternatively be mediated by such cells instead of bitter TRCs.

If the authors do not directly show that type II TRCs are involved, these alternative possibilities should be considered in their discussion.

Response: This is a reasonable point, and it was not our intention to make a strong statement as to the identity of the cells that mediate the Type III cell/OTOP1 independent/ Skn-1a dependent component of the behavioral aversion to NH₄Cl. We now discuss these other possible contributions to the behavioral aversion to NH₄Cl and remove any statements that could be misinterpreted to appear that there is any evidence that the Skn1a-dependent component of the behavioral response is mediated by bitter TRCs, which remains to be determined.

4. The abstract and introduction suggest that otopetrin is the primary means by which ammonium chloride is detected by the taste system. However, as noted above, their behavioral data suggest otherwise. The abstract and introduction should be rewritten to more clearly indicate that otopetrin is one mechanism by which ammonium chloride is detected by the taste system, but that there are others.

Response: we have made the edits the reviewer suggested. For example, the abstract now ends with “These data together reveal an unexpected role for the proton channel OTO1 in mediating a major component of the taste of NH₄Cl (underlined text was added)

Reviewers' Comments:

Reviewer #1:

Remarks to the Author:

The authors adequately addressed most of my previous concerns. However, let me suggest three minor modifications.

- (1) Line 380: "1-20" should probably be "5-20", according to Fig.1d. (this is my previous comment, but this has not been addressed in the revised manuscript.)
- (2) Fig. 1d: the unit of NH_4^+ should be indicated in the figure or legend. (this is my previous comment, but this has not been addressed in the revised manuscript.)
- (3) Line 375: Is "of e" necessary?

Reviewer #2:

Remarks to the Author:

Comments to authors

1) I appreciate the authors reply that the experiments to test the mechanism underlying the larger-than-expected currents are challenging, and may require substantial effort beyond the scope of this manuscript to interpret properly. Although the large currents observed could be due to OTO1 gating (by NH_4 or alkalization), the authors have not strictly ruled out other mechanisms, such as perhaps the channels themselves having some NH_4^+ permeability (and therefore contributing to the species-specific differences in H^+/NH_4^+ current ratios). The authors have made some textual changes to acknowledge that the mechanism underlying the currents and role of the identified amino acid is unclear:

Line 286, "It is presently unclear how changes in the intracellular milieu act on the arginine residue, or interacting partners, to increase the OTO1 currents."

Line 217 " NH_4Cl alkalizes the cell cytosol, creating a driving force for proton entry while at the same time acting through an intracellular arginine residue to increase current magnitudes, possibly by gating the channels."

However, I think the abstract needs to be more carefully written to avoid the impression that NH_4Cl gates the channel ("activate" reads similar to "gate"). For instance, in Line 5, "Here we report that OTO1, a proton channel expressed in sour (type III) taste receptor cells (TRCs), is activated by ammonium chloride (NH_4Cl), and required for its detection by Type III TRCs", the phrase "activated by ammonium chloride" should be omitted.

Likewise, I find that Lines 6-9 of the abstract are somewhat unclear. Maybe something like, "Extracellular NH_4Cl application leads to alkalization of the cell cytosol due to transit of NH_3 gas into the cell. In heterologous cells expressing OTO1, NH_4Cl -induced alkalization increases the driving force for proton entry through the OTO1 channel, leading to large inward currents that vary in magnitude in a species-specific manner. Sensitivity to ammonium requires a conserved intracellular arginine residue (R292) in the OTO1 tm 6-tm 7 linker.

2) I noticed the Liman group (Teng et al., Current Biology 2019) had previously reported that there was no difference in ammonium chloride responses in chorda tympani nerve recordings in OTO1 KO mice vs controls (although responses were partially reduced in the glossopharyngeal nerves)(Figure 5). In this manuscript, a key finding is that that ammonium chloride responses in chorda tympani nerves are nearly eliminated in OTO1 KO mice. Can the authors explain this significant difference?

Responses to reviewers:

Reviewer #1 (Remarks to the Author):

The authors adequately addressed most of my previous concerns. However, let me suggest three minor modifications.

(1) Line 380: "1-20" should probably be "5-20", according to Fig.1d. (this is my previous comment, but this has not been addressed in the revised manuscript.)

Response: That is correct. It has been changed.

(2) Fig. 1d: the unit of NH_4^+ should be indicated in the figure or legend. (this is my previous comment, but this has not been addressed in the revised manuscript.)

Response: The units are now in the figure legend.

(3) Line 375: Is "of e" necessary?

Response: This was a typo that has now been corrected.

We thank the reviewer for their careful review of the manuscript.

Reviewer #2 (Remarks to the Author):

Comments to authors

1) I appreciate the authors reply that the experiments to test the mechanism underlying the larger-than-expected currents are challenging, and may require substantial effort beyond the scope of this manuscript to interpret properly. Although the large currents observed could be due to OTOPI gating (by NH_4 or alkalization), the authors have not strictly ruled out other mechanisms, such as perhaps the channels themselves having some NH_4^+ permeability (and therefore contributing to the species-specific differences in H^+/NH_4^+ current ratios). The authors have made some textual changes to acknowledge that the mechanism underlying the currents and role of the identified amino acid is unclear:

Line 286, "It is presently unclear how changes in the intracellular milieu act on the arginine residue, or interacting partners, to increase the OTOPI currents."

Line 217 "NH₄Cl alkalizes the cell cytosol, creating a driving force for proton entry while at the same time acting through an intracellular arginine residue to increase current magnitudes, possibly by gating the channels."

However, I think the abstract needs to be more carefully written to avoid the impression that NH₄Cl gates the channel ("activate" reads similar to "gate"). For instance, in Line 5, "Here we report that OTOPI, a proton channel expressed in sour (type III) taste receptor cells (TRCs), is activated by ammonium chloride (NH₄Cl), and required for its detection by Type III TRCs", the phrase "activated by ammonium chloride" should be omitted.

Likewise, I find that Lines 6-9 of the abstract are somewhat unclear. Maybe something like, "Extracellular NH₄Cl application leads to alkalization of the cell cytosol due to transit of NH₃ gas into the cell. In heterologous cells expressing OTOPI, NH₄Cl-induced alkalization increases the driving force for proton entry through the OTOPI channel, leading to large inward currents that vary in magnitude in a

species-specific manner. Sensitivity to ammonium requires a conserved intracellular arginine residue (R292) in the OTOP1 tm 6-tm 7 linker.

Response: We appreciate the reviewer's effort to rewrite lines 6-9 in our abstract. To correct the grammar issues, and stay within the word limit we changed the abstract to use this wording:

"Here we report that OTOP1, a proton-selective ion channel expressed in sour (Type III) taste receptor cells (TRCs), functions as sensor for ammonium chloride (NH₄Cl). Extracellular NH₄Cl evoked large dose-dependent inward currents in HEK-293 cells expressing murine OTOP1 (mOTOP1), human OTOP1 and other species variants of OTOP1, that correlated with its ability to alkalinize the cell cytosol. Mutation of a conserved intracellular arginine residue (R292) in the mOTOP1 tm 6-tm 7 linker specifically decreased responses to NH₄Cl relative to acid stimuli. "

We provide multiple lines of evidence that the effect of ammonium is not due to simply the change in driving force for the proton (e.g. the mutation of an intracellular arginine residue, which could not change the driving force). The reviewer now raises the issue of whether NH₄⁺ itself is permeable through the channel. This is completely implausible as (1) these channels do not possess a pore that can accommodate and NH₄⁺ ion as we showed in CryoEM structures described in Saotome et al., 2019. Moreover, in Tu et al., 2018, we showed that the channels are not permeable to other smaller cations ions such as K⁺, Na⁺, Ca²⁺ (2) the reversal potential of the currents (Figure 3F) is not consistent with NH₄⁺ permeation (Our data show a reversal potential in response to 40 mM NH₄Cl of ~35 mV while the prediction assuming NH₃ (0.5 mM) crosses the membrane and re-equilibrates to generate 0.1 mM NH₄ is +153 mV).

2) I noticed the Liman group (Teng et al., Current Biology 2019) had previously reported that there was no difference in ammonium chloride responses in chorda tympani nerve recordings in OTOP1 KO mice vs controls (although responses were partially reduced in the glossopharyngeal nerves)(Figure 5). In this manuscript, a key finding is that that ammonium chloride responses in chorda tympani nerves are nearly eliminated in OTOP1 KO mice. Can the authors explain this significant difference?

Response: In Teng et al, 2019 we indeed found that the response of the glossopharyngeal nerve to NH₄Cl was mildly (but significantly) reduced in the *Otop1*^{-/-} mice but that the response of chorda tympani nerve to NH₄Cl was not significantly different between WT and *Otop1*^{-/-} mice. In these types of experiments, there is considerable variation due to the fact that we cannot normalize the data to a taste stimulus (typically this is NH₄Cl but sometimes in KCl), and this variability along with a small n may have obscured a finding of significance. To deal with the inherent variability in the data, in the present manuscript, we use data from CT nerve recordings from a much larger number of mice: 15 WT and 13 *Otop1*^{-/-} mice (Fig 1c).

We thank the reviewer for their input on the manuscript.